# Beta3-Tubulin Is Critical for Microtubule Dynamics, Cell Cycle Regulation, and Spontaneous Release of Microvesicles in Human Malignant Melanoma Cells (A375)

**DOI:** 10.3390/ijms21051656

**Published:** 2020-02-28

**Authors:** Mohammed O. Altonsy, Anutosh Ganguly, Matthias Amrein, Philip Surmanowicz, Shu Shun Li, Gilles J. Lauzon, P. Régine Mydlarski

**Affiliations:** 1Division of Dermatology, Department of Medicine, University of Calgary, Calgary, AB T2T 5C7, Canada; mohammedomarahmed.mo@ucalgary.ca (M.O.A.); ganutosh@umich.edu (A.G.); philip.surmanowicz@gmail.com (P.S.); glauzon@ualberta.ca (G.J.L.); 2Department of Zoology, Faculty of Science, Sohag University, Sohag 82524, Egypt; 3Department of Microbiology, Immunology and Infectious Diseases, University of Calgary, Calgary, AB T2N 4N1, Canada; shusli@ucalgary.ca; 4Department of Surgery, University of Michigan, Ann Arbor, MI 48105, USA; 5Department of Cell Biology and Anatomy, University of Calgary, Calgary, AB T2N 4N1, Canada; mamrein@ucalgary.ca

**Keywords:** melanoma, β3-tubulin, microtubules, microvesicles

## Abstract

Microtubules (MTs), microfilaments, and intermediate filaments, the main constituents of the cytoskeleton, undergo continuous structural changes (metamorphosis), which are central to cellular growth, division, and release of microvesicles (MVs). Altered MTs dynamics, uncontrolled proliferation, and increased production of MVs are hallmarks of carcinogenesis. Class III beta-tubulin (β3-tubulin), one of seven β-tubulin isotypes, is a primary component of MT, which correlates with enhanced neoplastic cell survival, metastasis and resistance to chemotherapy. We studied the effects of β3-tubulin gene silencing on MTs dynamics, cell cycle, and MVs release in human malignant melanoma cells (A375). The knockdown of β3-tubulin induced G2/M cell cycle arrest, impaired MTs dynamics, and reduced spontaneous MVs release. Additional studies are therefore required to elucidate the pathophysiologic and therapeutic role of β3-tubulin in melanoma.

## 1. Introduction

Class III beta-tubulin (β3-tubulin), encoded by the TUBB3 gene, is one of seven beta-tubulin isotypes in the human genome. It is constitutively expressed in neurons of the peripheral and central nervous system, regulating neuronal differentiation and development [1]. Microtubules (MTs), which comprise the cytoskeleton in almost all eukaryotic cells [2], are assembled through the dimerization of α- and β-tubulin proteins [3]. The dynamics of MTs, which involve phases of elongation, shortening, and pause [4], are relevant to intracellular trafficking, the formation of the mitotic spindle, cytokinesis, cell membrane blebbing, cell migration, and phagocytosis [2,5,6,7].

Microvesicles (MVs) are secreted by intact cells as microscopic membrane-enclosed sacs ranging in diameter from 100 to 1000 nm. They are formed by plasma membrane protrusions (buds), which then close and become separated from the progenitor cell, eventually to be released into the extracellular environment [8,9]. MVs are distinguished from the more homogeneous exosomes (40–100 nm), which are formed intracellularly within microvesicular bodies, and are thus of endosomal origin [10,11]. Membrane-bound proteins and luminal content determine the MV’s function, which consequently relates to the cell of origin [9,12]. Melanoma cell-derived MVs are postulated to play a biological role in the process of carcinogenesis. For example, malignant transformation of melanocytes is associated with the increased production and release of MVs with procoagulant activity, leading to a hypercoagulable state, a major cause of death in cancer patients [13,14]. Murine melanoma cell-derived MVs promote metastasis and immunosuppression through regulatory T-cell expansion and apoptosis of tumor-specific CD8^+^ cytotoxic T-cells, effects mediated by the MV-associated Fas ligand or TRAIL [15].

Furthermore, the remote delivery of peripheral mast cells originated-MVs to lymph nodes via the lymphatics would allow for more systemic modulation of inflammation and immunosuppression [10]. A similar mechanism may be involved in the invasion of melanoma cells into the dermis. Overall, MVs are postulated to promote cancer initiation, survival, and spread [12,13,16]. Herein, we hypothesize that β3-tubulin interferes with microtubule dynamics, cell cycle regulation, and the spontaneous release of MVs in A375 human malignant melanoma cells. 

## 2. Results

### 2.1. A375 Cells Express β3-Tubulin mRNA and Protein

β3-tubulin is highly expressed in malignant as compared with normal melanocytes, and siRNA knockdown of β3-tubulin hinders the migration of melanoma cells [17]. Using RT-PCR and Western blot analysis, we demonstrated that the A375 human malignant melanoma cells express β3-tubulin mRNA and protein (Figure 1A). siRNA knockdown of β3-tubulin significantly reduced the mRNA and protein expression by 79.3% ± 7.7% (*p* < 0.001) and 83% ± 4.9% (*p* < 0.01), respectively. The effect of treatment with either β3-tubulin siRNA or control siRNA on the immunostaining of melanoma cells A375 with both β3-tubulin and a specific molecular marker for melanoma cells (melanoma-associated antigen, or MAA) confirmed the specificity of β3-tubulin knockdown by β3-tubulin siRNA (Figure 1B and Appendix A), with no observed effect on cell viability assessed by an MTT assay (Figure 1C). 

### 2.2. siRNA Knockdown of β3-Tubulin Reduces the Spontaneous Release of MVs by A375 Cells

The spontaneous release of MVs by A375 cells was assessed using Alexa fluor 488-labeled wheat germ agglutinin (WGA) to stain cell membranes, followed by fluorescence microscopy. A375 cells release ring-like MVs with clear intravesicular spaces with sizes ranging from 200 to 1000 nm (Figure 1F). MVs were also detected in the culture medium (Figure 1G). The effect of β3-tubulin knockdown on MVs counts was assessed by flow cytometric quantification of MAA-labeled MVs in the culture media of naïve, control-, and β3-tubulin siRNA-treated cells. Knockdown of β3-tubulin induced a significant 48.63% (*p* < 0.001) reduction of released MVs, relative to naïve cells (Figure 1D,E), while control siRNA did not affect MVs release from cells. These data demonstrate that β3-tubulin modulates, at least in part, spontaneous MVs release from A375 melanoma cells.

### 2.3. siRNA Knockdown of β3-Tubulin Suppresses MTs Dynamics and Induces G2/M Cell Cycle Arrest

Altered cellular MTs dynamics, including elongation, shortening, and a pause, are salient to carcinogenesis, broad-spectrum chemotherapy resistance and cell survival [18]. The effect of β3-tubulin knockdown on MTs dynamics in A375 cells transfected with EGFP-microtubule-associated protein-4 cDNA (EGFP-MAP4) was assessed. A sequence of frames, five-second apart, of the life-history of MTs +ends transitioning between phases of elongation (growth), shortening, and pause is shown (Figure 2A). MTs +end displacements (growth/shortening) of naïve and control siRNA-treated cells were greater than those of β3-tubulin knockdown cells (Figure 2A, Appendix A). The rate of total growth and total shortening were reduced by 39.7% ± 5% (*p* < 0.001) and 52% ± 4.7% (*p* < 0.001), respectively, in β3-tubulin knockdown cells as compared to naïve cells, whereas control siRNA treatment had no effect on these parameters. Further, pause frequency was significantly higher in β3-tubulin knockdown cells by 19.04 % ± 4.2% (*p* < 0.05), whereas overall MTs dynamicity was significantly reduced by 46.53% ± 4.3% (*p* < 0.001).

Cell cycle analysis revealed that growth phase redistribution in β3-tubulin knockdown cells as compared to naïve cells, where G2/M and polyploidy (> G2/M) cells were significantly increased by 53.12% (*p* < 0.001) and 245.74% (*p* < 0.001), respectively. Furthermore, G0/G1 cells were significantly reduced by 45.49% (*p* < 0.001). Similar effects were not seen in control siRNA-treated cells. There was no significant change in the less than G0 (< G0) and S populations (Figure 2C and Appendix A).

## 3. Discussion

By understanding the mechanisms of carcinogenesis at the cellular and molecular level, current research focuses on targeting strategies for melanoma prevention and treatment. With essential regulatory roles in a myriad of inter- and intra-cellular activities, MTs dynamicity is an important target for many anti-cancer therapeutics [19,20,21,22]. β-tubulin is a structural protein that maintains the microtubule cytoskeleton. Distinct from the other six beta-tubulin isotypes in the human genome, β3-tubulin has a unique molecular structure that facilitates its binding to factors involved in the oxidative stress and nutrient deprivation response, thereby bolstering the survival of stressed cells [23]. For example, induced expression of β3-tubulin promotes resistance to the widely used chemotherapeutic agent, paclitaxel, in Hela and MCF-7 cell lines [24]. Furthermore, increased levels of β3-tubulin expression have been linked to malignant transformation and migration of melanocytes [17]. In tandem with these studies, our Q-PCR, Western blot, and immunofluorescence data confirmed that β3-tubulin was highly expressed in the human malignant melanoma cell line (A375).

To understand the relevance of β3-tubulin on melanoma progression, we used siRNA to knockdown β3-tubulin in the A375 cell line. We studied several factors known to contribute to metastasis and drug resistance, namely MVs release, MTs dynamics, and cell cycle regulation [19,20,21,22,25]. Our data demonstrated significant suppression of MTs dynamic parameters (e.g., growth, shortening, and pause frequencies), implying that β3-tubulin is critical for MTs dynamicity. As normal MTs dynamics are essential for mitotic spindle formation and function, disrupting MTs dynamics would lead to the activation of the cell cycle checkpoints and force the malignant cells to initiate growth arrest and apoptotic cell death [26]. Despite the importance of β-tubulin isotypes in resistance to antimitotic drugs [27,28], there are no studies to date examining the role of β3-tubulin in cell cycle regulation in malignant melanoma. Thus, we investigated the effect of β3-tubulin siRNA on the cell cycle in A375 melanoma cells, and the data demonstrated robust induction of G2/M arrest and a significant increase in polyploidy population in β3-tubulin siRNA-treated cells. Such data provide direct evidence that β3-tubulin alters melanoma cell cycle regulation. G2/M arrest, which may be the result of defective mitotic spindle formation, was shown to enhance the cytotoxic effect of chemotherapy in melanoma cell lines [29,30]. These results are consistent with previous work on Hela cells [31], where the reduction of β3-tubulin expression by siRNA resulted in partial inhibition of cell growth. Interestingly, in H460 (non-small cell lung cancer), β3 tubulin siRNA increased vincristine-and paclitaxel-induced suppression of microtubule dynamicity and cell death at low and high drug concentrations, respectively, while there was no effect on the G2/M population [32]; such data might propose varied and cell-specific roles of β3 tubulin in different cell types.

Despite the importance of MVs as potential therapeutic targets in cancer (i.e., angiogenesis, matrix degradation and invasion, metastasis, and immunosuppression) and thrombosis [9,12,33], the mechanisms of their biogenesis and release remain poorly characterized. Several lines of evidence demonstrate that MVs are important for melanoma progression [13,14], coagulation [12], and inflammatory skin conditions (i.e., psoriasis) [34]. The correlation between MTs dynamics and MVs release was previously studied in mastocytoma P815 cell [35], where the authors demonstrated that microvesicle formation and shedding require MTs disruption. The role of MTs in controlling other cellular activities, such as movement and intracellular vesicle transportation was also reported [36,37]. However, the mechanisms of MVs biogenesis and release in melanoma cells remain controversial. Herein, we demonstrated that β3-tubulin knockdown-induced repression of MTs dynamicity was associated with a reduction in the numbers of the MVs released by A375 cells, implying that β3-tubulin may play a role in MVs production via regulating the MTs dynamics in melanoma cells. Further studies are required to elucidate the mechanistic role of the α- and β- tubulins isotypes in the treatment and prevention of melanoma progression.

## 4. Materials and Methods

### 4.1. Cell Culture EGFP-MAP-4 Transfection and siRNA Silencing

A375 human malignant melanoma cells (ATCC^®^ CRL-3224) were propagated in Dulbecco′s Modified Eagle′s Medium (DMEM—catalog no. 11995-065, Life Technologies, Grand Island, NY, USA) containing 4.5 g/L D-glucose, L-glutamate, 110 mg/L sodium pyruvate, and supplemented with 10% fetal bovine serum (FBS—catalog no. 098105, MULTICEL) and 1% penicillin-streptomycin (catalog no. 15140-122, Life Technologies). EGFP-MAP-4 transfection (provided by Dr. Joanna Olmsted, University of Rochester, USA) was performed as previously described [24]: briefly, cells were seeded in twelve-well plates at a density of 150,000 cells per well and transfected with 1 μg plasmid DNA using lipofectamine 2000 (catalog no. P/N52887, Invitrogen, Carlsbad, CA, USA) following the manufacturer’s instructions. Transfected cells were selected by growing in a growth medium containing G418, 2 mg/mL (catalog no. A1720; Sigma-Aldrich, Oakville, ON, Canada). For siRNA silencing, A375 cells were transfected with 25 nM of β3-tubulin siRNA (catalog no. sc-105009, Santa Cruz Biotechnology, Dallas, TX, USA) or FlexiTube Lamin A/C non-targeting siRNA (catalog no. SI03650332, Qiagen, Valencia, CA, USA), using transfection reagent lipofectamine RNAiMAX (catalog no. 13778-075, Thermo Fisher Scientific, Carlsbad, CA, USA) as per the manufacturer’s protocol. Western analysis, RNA isolation, immunofluorescent microscopy, and MVs purification were performed 48 h after transfection, unless otherwise stated.

### 4.2. RNA Isolation, cDNA Synthesis, and qPCR

Total RNA was extracted using NucleoSpin RNA purification kits (catalog no. 740955- 250, D-Mark Biosciences, Düren, Germany) and 500 ng was used for cDNA synthesis (qScript cDNA Synthesis kit, catalog no. CA101414-098, Quanta Biosciences, Gaithersburg, MD, USA). Real-time PCR (RT-PCR) was performed with a StepOne Plus PCR machine (Applied Biosystems, Foster City, CA, USA) using the fast SYBR Green master mix (catalog no. 4385618, Life Technologies). The amplification conditions were 95 °C for 20 s followed by forty cycles at 95 °C for 3 s and 60 °C for 30 s. Primer sequences were as follows: β3-tubulin, forward 5′-CGA AGC CAG CAG TGT CTA AA-3′, reverse 5′-GGA GGA CGA GGC CAT AAA TAC-3′; ribosomal protein L19 (RPL19), forward 5′-ATC GAT CGC CAC ATG TAT CA-3′, reverse 5′-GCG TGC TTC CTT GGT CTT AG-3′.

### 4.3. Electrophoresis, Western Analysis, and Fluorescent Microscopy

Fifty micrograms of A375 melanoma cell lysate protein in radioimmunoprecipitation assay buffer (RIPA) containing 1% of Halt™ protease inhibitor cocktail (catalog no.1861279, ThermoScientific) was separated by SDS-PAGE and transferred onto nitrocellulose membranes (catalog no. rpn203d, GE Health). Membranes were immunoprobed with rabbit monoclonal anti-human β3-tubulin (catalog no. d71g9-xp, Cell Signaling), or mouse anti-human glyceraldehyde-3-phosphate dehydrogenase (GAPDH; catalog no. 4699-9555, Biogenesis). The secondary antibodies were peroxidase-conjugated goat anti-rabbit IgG (catalog no. 111-035-003, Jackson ImmunoResearch) and peroxidase-affini-pure goat anti-mouse IgG (catalog no. 115-035-003, Jackson ImmunoResearch). Immune complex visualization was carried out using ECL TM prime WB reagents (catalog no. rpn2232sk, GE Health). For fluorescence microscopy, A375 melanoma cells were fixed in pre-warmed culturing medium containing 3.7% formaldehyde (catalog no. f-8775, Sigma) for 15 min at 37 °C. Cellular and vesicular membranes were labeled with Alexa 488-conjugated wheat germ agglutinin (WGA, catalog no. W11261, Invitrogen) 5 μg/mL diluted in PBS for 10 min at room temperature followed by two washes in PBS. For cytoplasmic and inner membrane-bound protein immuno-detection, cells were permeabilized for 5 min at room temperature with 0.2% tween X-100 (catalog no. t9284, Sigma) in PBS and blocked for 1 h at room temperature using a blocking buffer (PBS containing 5% albumin bovine serum, catalog no. a-4503, Sigma). The cells were probed with antibodies against β3-tubulin, melanoma associated antigen (MAA, mouse monoclonal anti human, catalog no. ab34165, Abcam), isotype controls (rabbit IgG (monoclonal, catalog no. ab172730, Abcam), or mouse IgG 2b (catalog no. 557351, BD Pharmingen)) diluted in blocking buffer and incubated overnight at 4 °C. The secondary antibodies used were rabbit Alexa fluor 488 goat anti-rabbit (catalog no. A11034, Invitrogen) and mouse Alexa fluor 555 goat anti-mouse (catalog no. A21424, Life technologies). 4′,6-diamidino-2-phenylindole (DAPI, catalog no. d21490, Molecular probes) was used as a nuclear counterstain marker. Immunoprobed cells were mounted using prolong gold anti-fad reagent (catalog no. p36930, Invitrogen) and visualized by confocal microscopy and structured illumination super-resolution microscopy (Zeiss Elyria).

### 4.4. MVs Purification and Flowcytometric Quantification

MVs purification was carried out following the referenced protocol by Lima et al. [13]. Briefly, A375 melanoma cell culture supernatants were centrifuged at 800× *g* for 10 min to exclude floating dead cells and debris. Another centrifugation for 15 min at 14,000× *g* yielded a pellet, which was resuspended and recollected by repeat centrifugation (this and subsequent centrifugations were at 14,000× *g* for 15 min at 4 °C). The washed pellet containing MVs was suspended in 100 μL ice-cold PBS. MVs were fixed by adding 900 μL of 100% methanol at −20 °C dropwise with vigorous vortexing. This suspension was incubated at −20 °C for 5 min followed by the addition of 1 mL of ice-cold PBS. Fixed MVs were pelleted by centrifugation, resuspended in 250 μL blocking buffer and incubated at room temperature for 1 h. MVs were collected by centrifugation and immunostained overnight at 4 °C in 250 μL of the blocking buffer containing a mouse monoclonal MAA antibody or mouse IgG 2b isotype control at a concentration of 1 μg/mL. MVs were then washed once in PBS, incubated one hour at room temperature in a blocking buffer containing Alexa fluor 488 goat anti-mouse (catalog no. a11001, Life Technologies) and washed once in PBS. MVs quantification was assessed using a flow cytometry (Guava^®^ easyCyte; MilliporeSigma). Data analysis was carried out using FCS express 6 plus research edition software.

### 4.5. Analysis of Microtubules Dynamics

A375 cells were transfected with EGFP-MAP-4 (provided by Dr. Joanna Olmsted, University of Rochester, USA) as previously described [24]. MTs dynamics analysis was performed as previously published [38]: briefly, MTs +ends were tracked using ImageJ software (Manual tracking plugin, https://imagej.nih.gov/ij/plugins/track/track.html). Only 0.5 μm changes in growth/shortening traces were plotted for each MT. Any growth or shortening event below the 0.5 μm threshold was considered a pause. The growth/shortening rate was calculated by dividing the sum of all growth (positive) or the sum of all shortening (negative) by the total time spent growing or shortening respectively. Pause frequencies were calculated by dividing the total number of pauses by the total measuring time. MTs dynamicity was extracted by dividing the sum of total length (growth and shortening) by the total measuring duration. Data was analyzed using GraphPad Prism 6, and MS-Excel software. Data Significances were calculated using a student *t*-test and one-way analysis of variants (ANOVA) with Tukey’s correction for multiple comparison.

### 4.6. Propidium Iodide (PI)-Staining and Cell Cycle Analysis

A375 cells were fixed for 30 min in 70% ethanol at 4 °C and washed 2× in PBS. RNase (100 μg/mL) was added, and the cells were incubated for 20 min at 37 °C, followed with 2× washes in PBS. Cells were then incubated in 3 μM PI (catalog no. p4170; Sigma) in the staining solution (100 mM tris, PH 7.4, 150 mM NaCl, 1 mM CaCl_2_, 0.5 mM MgCl_2_, and 0.1% Nonidet P-40) for 15 min at room temperature. Cell cycle analysis was conducted using a flow cytometer (Guava^®^ easyCyte; MilliporeSigma). Data analysis was carried out using FlowJo v10.6.1 software.

## 5. Conclusions

β3-tubulin knockdown suppresses MTs dynamics, decreases MVs release, and induces G2/M cell cycle arrest in human malignant melanoma cells (A375). β3-tubulin, an important microtubular protein, may therefore be used to study the role of MTs and MVs in the pathogenesis and treatment of melanoma.

## Figures and Tables

**Figure 1 ijms-21-01656-f001:**
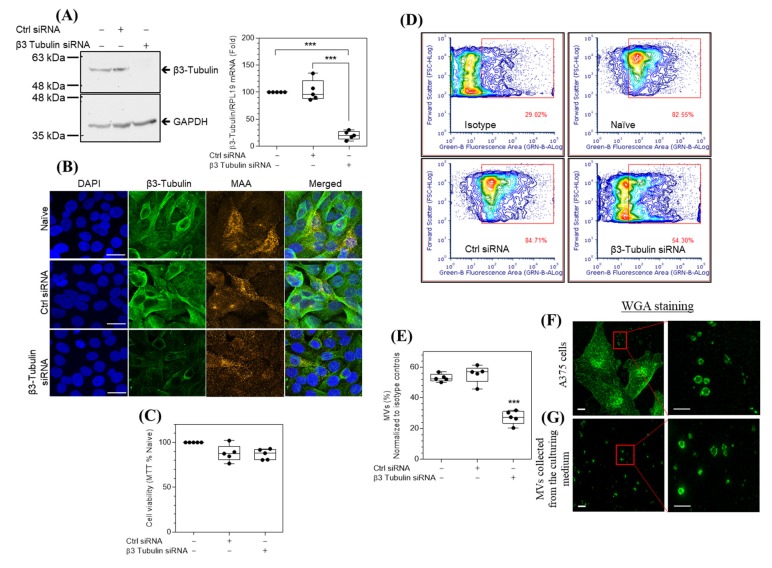
β3-tubulin knockdown reduces the numbers of microvesicles (MVs) spontaneously released by A375 melanoma cells under the standard culture conditions. (**A**) Western blots of β3-tubulin and GAPDH loading control (panel A, left), and RT-PCR quantification of β3-tubulin mRNA expression normalized to the C_T_ values of RPL19 (panel A, right). (**B**) Naïve, control (ctrl)−, and β3-tubulin siRNA-treated cells were immunostained with antibodies specific for β3-tubulin (green) and MAA (orange), DAPI (blue) was used as nuclear DNA counterstain, original magnification ×630 (bar, 20 μm). (**C**) MTT assay shows the effect of siRNA treatment on the cellular viability of A375 cells. Flow cytometric contour plots (**D**) and quantification (**E**) of MVs collected from the culture media of naïve, ctrl-, and β3-tubulin-siRNA treated cells and immunostained with an MAA antibody or isotype control, plots illustrate the percentage of positively stained MVs (D, red square gated events). Super-resolution microscopy with maximum intensity projection of a confocal stack micrographs of A375 cells (**F**), or MVs collected from culture media (**G**), stained with Alexa fluor 488-conjugated-WGA, red rectangles, in F and G, show randomly selected areas for higher magnification (right enlarged micrographs), illustrating ring-like microvesicles of varying sizes (200–1000 nm) with optically clear lumen. Original magnification ×1000, scale bar 5 μm (left graphs) or 1 μm (magnified right graphs). Statistical significance was determined between different groups using an ANOVA with Tukey’s correction for multiple comparisons. *n* = 5, *** *p* < 0.001 versus naïve cells.

**Figure 2 ijms-21-01656-f002:**
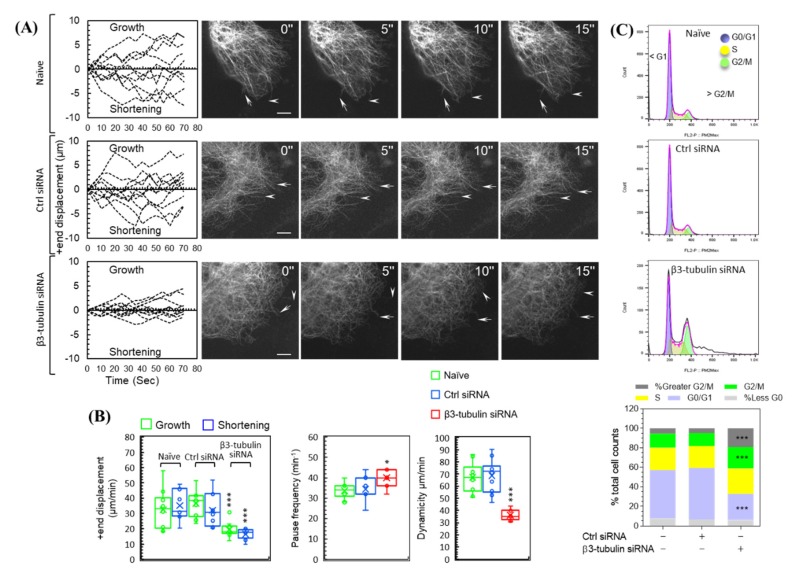
β3-tubulin knockdown suppresses microtubules (MTs) dynamics and disrupts cell cycle in A375 cells. (Panel (**A**), right) Micrographs of EGFP-MAP-4-labeled MTs in naïve, ctrl-, and β3-tubulin siRNA-treated A375 cells. Images were acquired every 5 s and show MTs +end growth (arrows) and shortening (arrowheads). Original magnification ×1000 (bar, 5 μm). (panel (**A**) left) Traces of ten representative MTs +end displacements (growth and shortening) measured over a period of 75 s. Quantification of dynamic growth and shortening (panel (**B**), left), and pause frequency (panel (**B**), middle) and dynamicity (panel (**B**), right). (**C**) Flow cytometric histograms (top panels), and cell cycle growth phase distribution analysis (bottom panel) demonstrate the effect of β3-tubulin knockdown on cell cycle in A375 cells. Statistical significance between naïve, ctrl-, and β3-tubulin-siRNA groups was determined using an ANOVA with Tukey’s correction for multiple comparisons. *n* = 10 (MTs analysis) or = 6 (cell cycle analysis), *** *p* < 0.001, and * *p* < 0.05 versus naïve cells.

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
