# Peer review of "Beta3-Tubulin Is Critical for Microtubule Dynamics, Cell Cycle Regulation, and Spontaneous Release of Microvesicles in Human Malignant Melanoma Cells (A375)"

_ijms, 2020, doi:10.3390/ijms21051656_

Round 1
Reviewer 1 Report
In this version of the manuscript, the claims were supported by the experimental data.
Author Response
Response: The authors thank the reviewer’s positive response to our revision.
Reviewer 2 Report
The Authors added flow cytometry data showing the effect of the depletion of beta-3 tub in A375 cells on the cell-cycle profile. The discussion section is still mostly the repetition of the results section. As indeed the data on the role of beta-3 tub in melanoma cells are limited, Authors could discuss how siRNA-based knockdown of beta-3 tub affects MTs dynamics and cell-cycle profile in other cells. This work still needs to be improved.
Reviewer: 2
- Reviewer #2: The submitted manuscript “Beta3-tubulin knockdown suppresses microtubule dynamics and decreases the spontaneous release of microvesicles in human malignant melanoma cells (A375)” provides rather preliminary data. It shortly describes simple observations that knockdown of beta3 tubulin changes dynamics of the microtubules and that such cells release fewer microvesicles. Regretfully, the Authors failed to provide a hypothesis of the molecular mechanism(s) behind such changes. The production of MVs is increased in cancer or damaged cells possibly due to the metabolic re-programming of these cells. Are there any data showing a direct connection between MTs and MVs formation? Or is this an indirect effect caused by the changes in the cell’s program?
Authors response: The authors have conducted additional mechanistic studies. Despite the importance of β-tubulin isotypes in resistance to antimitotic drugs, there are no studies to date which examine the role of 3-tubulin in malignant melanoma. Herein, the authors use flow cytometry to investigate the effect of β3-tubulin siRNA on cell cycle distribution in A375 melanoma cells. The robust induction of G2/M arrest and the significant increase in polyploidy of β3-tubulin siRNA-treated cells provides direct evidence that β3-tubulin alters melanoma cell cycle regulation. Furthermore, G2/M arrest, which may be the result of defective mitotic spindle formation, enhances the cytotoxic effect of chemotherapy in melanoma cell lines.
Reviewer #2 – 12 Feb 2020
The additional data showing the effect of the beta3 knockdown on the cell cycle of A375 cells improve this manuscript. However, this interesting piece of data were not properly discussed in the Discussion section.
Moreover, in the previous comment, I asked also how beta-3 tub (and microtubule dynamics) can affect MV formation. In the discussion section Authors stated:
“In melanoma progression, MVs are associated with increased procoagulant activity, immunosuppression, inflammation, and metastasis [13, 14, 33]. Furthermore, MTs control cellular movements and vesicle transportation [34, 35]. However, the mechanisms of MV biogenesis and release remain controversial.”
Placing the information that: “ Furthermore, MTs control cellular movements and vesicle transportation [34, 35]” (that is true for the cytoplasmic vesicles), between two sentences about microvesicles, makes an impression that MTs are involved in microvesicle transport. However, microvesicles bud off directly from the plasma membrane. So this part has to be corrected.
If nothing about mechanism of MV in melanoma cells is known, Authors should cite papers that address this issue in other cell lines (suggesting possible involvement of actin). The question remains if knockdown of beta3-tub causing reduced dynamics of the microtubules affects subcortical skeleton / actin filaments or cause general alterations in the cell [Authors have cited Cicchillitti et al., 2008] that indirectly affect formation of the MV. Please discuss this issue.
- Reviewer #2: If MTs are formed and beta3 tub is reduced, which isoform of beta-tubulin substituted beta3-tub?
Authors response: As β3-tubulin is one of the major MT structural proteins, its relative reduction significantly altered the MTs assembly and disassembly processes (elongation and shortening). These data addressed our hypothesis that β3-tubulin mediates MT dynamicity and ultimately reduces the cell’s capacity to release MV. Furthermore, β3-tubulin siRNA-treated A375 cells were still expressing the protein (see immunofluorescence, Figure 1B & Supplemental figure S1) as the siRNA would not completely knockout β3-tubulin expression, so that we would be able to detect it, which may explain the continuation of the formation of, somewhat less functional, MT.
Reviewer #2 – 12 Feb 2020
In cells transfected with beta3-tub siRNA the level of beta-3 tub was reduced to about 20%. Microtubules are composed of a- and b-tubulin, so if “β3-tubulin is one of the major MT structural proteins” logically, one should see a significant reduction in the microtubular network. For this reason, it was suggested to stain cells with anti-a-tubulin antibodies, to see the entire microtubular cytoskeleton in the cell, not only to know what is happening at the microtubules ends (elongation, shortening and pauses). Based on Fig 2A (EGFP-MAP4) MTs in beta-3 tub depleted cells seems to be more dense (?). Is the microtubular mass higher in these cells?
The genome encodes several b-tubulin isotypes. One can’t exclude that such decrease of the level of one isotype can be compensated by the other beta isotype. Please see Figure 1 in Gan et al., 2010, (Microtubule Dynamics, Mitotic Arrest, and Apoptosis: Drug-Induced Differential Effects of βIII-Tubulin).
In the same publication Authors can find how knockdown of beta-3- tub affects microtubule dynamics in Human NSCLC H460 cells (non-small-cell lung cancer) - (Table 1 – nontreated samples) and address in the discussion section how knockdown of beta-3- tub affects MT dynamics in NSCLC. They can also discuss if the outcome of beta3 depletion can be cell-type specific. Are there any more studies (using other cell types as a model) showing how depletion of beta-3 tub affects MT dynamics?
- Reviewer #2: Fig 1. Please provide large immunofluorescence images of the control and knockdown cells so one could clearly see changes.
Authors response: Larger immunofluorescence images of the control and knockdown cells were added to the supplementary data file (Supplemental figure S1).
Reviewer #2 – 12 Feb 2020
Thank you, please cite Figure S1 in the main text.
- Reviewer #2: Figure 2 is missing (?)
Authors response: The authors believe that the absence of figure 2 from the main text was due to a technical error. They have sent a note to the editorial assistant team to make sure that all figures are included in the text.
Reviewer #2 – 12 Feb 2020
In the previous version of the manuscript the missing Fig 2 was not cited J
- Reviewer #2: According to Shibazaki et al., 2012, in in vitro cultured cancer cells, constitutive TUBB3 expression seems to be regulated in a cell-cycle dependent way, with maximal expression at the G2/M phase of the cell cycle [Shibazaki M, et al., Transcriptional and post-transcriptional regulation of beta III-tubulin protein expression in relation with cell cycle-dependent regulation of tumor cells. International journal of oncology 2012, 40(3):695–702.]. Thus it would be of interest to investigate how knockdown of beta3 tubulin affects cell cycle (proliferation time, spindle morphology).
Authors response: The effect of β3-tubulin knockdown on the cell cycle regulation was studied. The results demonstrate that β3-tubulin knockdown induces G2/M arrest and a significant increase in polyploidy as compared to naïve- or control siRNA-treated cells.
Reviewer #2 – 12 Feb 2020
The reviewer appreciate the addition of the interesting piece of data (flow cytometry data). Use of DAPI instead of PI would make RNAse treatment unnecessary. Regretfully, in the discussion section Authors only repeat the information about the obtained results and suggest that:” Further, G2/M arrest, which may be the result of defective mitotic spindle formation, enhances the cytotoxic effect of chemotherapy in melanoma cell lines [31, 32]. Well, the assumption of the defective mitotic spindle can be easily addressed by simple immunofluorescence analysis. Have Authors seen such cells among cells transfected with beta-3 tub siRNA? In H460 cells knockdown of beta-3 tub did not affect cell cycle profile. Can Authors comment on this in their discussion?
- Reviewer #2: According to the Authors (line 135) beta 3 tub has a unique molecular structure. Beta 3 tubulin differ from other beta tubulins by several amino acids in the globular part (1-429, including a region of the paclitaxel binding domain) and by the C-terminal end (430-450) the place of the several tubulin posttranslational modifications. How knockdown of Beta3-tub changed resistance to paclitaxel and level of tubulin posttranslational modifications (e.g. acetylation and glutamylation?)
Authors response: The data suggest that β3-tubulin knockdown suppresses microtubule dynamics and decreases the spontaneous release of microvesicles in human malignant melanoma cells (A375). To highlight the importance of β3-tubulin in mediating MT dynamics, cell migration, and drug resistance in other cancer cell types, the authors discuss previously published work that examines the relationship between β3-tubulin expression and resistance to paclitaxel. Though interesting and clinically relevant, the proposed additional studies are beyond the scope of this manuscript.
Reviewer #2 – 12 Feb 2020
The goal of the proposed experiment (IF studies) was to fill a “gap” between the simple observations and the molecular mechanism behind the observed changes.
Additional comments:
To silence beta3 tubulin Authors used siRNA (catalog no. sc-105009,Santa Cruz Biotechnology) and as a control FlexiTube Lamin A/C non-targeting siRNA (catalog no. SI03650332,Qiagen). I was wondering (this information in missing in the Material and Methods section), how Authors distinguished between transfected and non-transfected cells. Were all cells transfected (100% transfection efficiency)? How was estimated the efficiency of the transfection? Was any marked added? If so, one should provide the parallel images showing which cells (in case of both, beta-III tubulin siRNA and control siRNA) were transfected and added information in the material and methods section.
The western blot analysis shows the tendency in the entire population (significant reduction of the level of beta-3 tub), however does not show what is happening in the individual cells. And in case of the MT dynamics Authors analyze individual cells and they have to be sure (and also convince Readers) that they analyze cell that was transfected and has reduced level of beta-3 tubulin.
Regarding Fig 1. To some extent the intensity of signal will also depend if one see an individual cells or overlapping cells. For example in naïve sample, cells in the center have stronger signal compared to cells in the image periphery and these in the periphery (naïve) have signal similar to the one observed in beta-3 tub why cells transfected with control siRNA has stronger beta-3 signal compared to naive cells.
In case of EGFP-MAP4 I could not find the information about the concentration used in the experiment and in case of silenced cells, if both siRNA and EGFP-MAP4 were co-transfected (in case of EGFP-MAP transfected cells can be distinguished based on EGFP signal).
How Authors verified that used beta-3 tub siRNA targets only beta-3 isotype?
Round 2
Reviewer 2 Report
The manuscript was corrected and most of the questions and issues raised by the reviewer were addressed
This manuscript is a resubmission of an earlier submission. The following is a list of the peer review reports and author responses from that submission.
Round 1
Reviewer 1 Report
This manuscript identified that Beta3-tubulin knockdown suppresses microtubule dynamics and decreases the spontaneous release of microvesicles (MVs) in human malignant melanoma cells. However, more sufficient information is needed before this manuscript can be accepted for publication.
Here are a few additional major suggestions.
1 In Figure 1B, there may be a change in spatial pattern of MAA after beta3-tubulin silencing. The authors should classify it to explain its subcellular localization more clearly.
2 I was confused a little bit from the gate of Isotype in the Figure 1 D. The authors showed that the MVs ranging from ∼200 to 1000 nm in diameter were obtained. It would be better to use a 0.22-μm pore size filter to remove exosomes and free proteins.
3 In Figure 1G and 1H, the authors should add scale bar to the representative figures with high-resolution to determine the sizes of MVs present within these preparations.
4 It would be better to conduct control experiments to demonstrate that the MV isolation procedure efficiently removed exosomes, cytosolic contamination and freely secreted proteins. For example, if the authors add one step to pass through the CM with a 0.22-μm pore size filter. The filter only retained vesicles larger than 0.22 μm in diameter (that is, the MVs), while the smaller exosomes as well as soluble factors will not be retained.
5 To strengthen the rationale, it would be better to add one more cell line to confirm the release of MVs upon beta3-tublin silencing.
Author Response
Reviewer: 1
Reviewer #1: In Figure 1B, there may be a change in spatial pattern of MAA after beta3-tubulin silencing. The authors should classify it to explain its subcellular localization more clearly.Author response: The melanoma associated antigen (MAA), a specific marker expressed on the surface of melanoma cells, was used to confirm the melanocytic origin of the A375 cell line. While there may be changes in the spatial pattern of MAA after β-tubulin silencing, this work is beyond the scope of a short communication. The differences in the fluorescence intensity between naïve, control-, and β3-tubulin siRNA-treated cells may be due to the minimal‐toxicity effect of siRNA-transfection (illustrated in figure 1C).
Reviewer #1: I was confused a little bit from the gate of Isotype in the Figure 1 D. The authors showed that the MVs ranging from ∼200 to 1000 nm in diameter were obtained. It would be better to use a 0.22-μm pore size filter to remove exosomes and free proteins.
Author response: Microvesicles (MV) are secreted by intact cells as microscopic, membrane-enclosed sacs ranging in diameter from 100-1000 nm (page no. 2, line no. 39 & 40). The use of 0.22 μm filters (≈ 220 nm) would exclude MVs smaller than the filter diameter, resulting in biased data. Purification of MVs from cell culture supernatant was carried out following the consecutive centrifugation method, a standard protocol referenced in the literature (references [1-3]). We have updated the manuscript’s reference list accordingly (page no. 8; line no. 202).
Lima, L.G., et al., Tumor-derived microvesicles modulate the establishment of metastatic melanoma in a phosphatidylserine-dependent manner. Cancer Lett, 2009. 283(2): p. 168-75. Muhsin-Sharafaldine, M.R., et al., Procoagulant and immunogenic properties of melanoma exosomes, microvesicles and apoptotic vesicles. Oncotarget, 2016. 7(35): p. 56279-56294. Martinez-Lorenzo, M.J., et al., Activated human T cells release bioactive Fas ligand and APO2 ligand in microvesicles. J Immunol, 1999. 163(3): p. 1274-81. Reviewer #1: In Figure 1G and 1H, the authors should add scale bar to the representative figures with high-resolution to determine the sizes of MVs present within these preparations.
Authors response: The scale bars have been added.
Reviewer #1: It would be better to conduct control experiments to demonstrate that the MV isolation procedure efficiently removed exosomes, cytosolic contamination and freely secreted proteins. For example, if the authors add one step to pass through the CM with a 0.22-μm pore size filter. The filter only retained vesicles larger than 0.22 μm in diameter (that is, the MVs), while the smaller exosomes as well as soluble factors will not be retained.
Authors response: As previously mentioned, the detection and quantification of MV release from A375 cells was carried out by staining the cells or the collected MVs using wheat germ agglutinin (WGA), which binds glycoproteins at specific melanoma binding sites, or MAA, which targets chondroitin sulphate proteoglycans expressed on the surface of melanoma cells. Any secreted proteins, smaller molecules or other cellular components that do not stain for WGA or MAA cannot be detected by either fluorescent microscopy or flow cytometry.
Reviewer #1: To strengthen the rationale, it would be better to add one more cell line to confirm the release of MVs upon beta3-tublin silencing.
Authors response: The literature suggests that β3-tubulin is largely responsible for melanoma cell migration (reference 17). Our work clearly demonstrates that β3-tubulin is important in microtubule dynamics, cell cycle regulation, and the spontaneous release of microvesicles in A375 cells, a well-established melanoma cell line. We present data to demonstrate that β3-tubulin knockdown induces G2/M arrest, suppresses microtubule dynamics and decreases the spontaneous release of microvesicles in human malignant melanoma cells. As A375 is a well-established human melanoma cell line, we believe that its use is sufficient for this proof-of-concept study. The authors believe that additional studies using different cell lines will significantly delay the dissemination of their important findings.
Reviewer 2 Report
The submitted manuscript “Beta3-tubulin knockdown suppresses microtubule dynamics and decreases the spontaneous release of microvesicles in human malignant melanoma cells (A375)” provides rather preliminary data. It shortly describes simple observations that knockdown of beta3 tubulin changes dynamics of the microtubules and that such cells release fewer microvesicles.
Regretfully, the Authors failed to provide a hypothesis of the molecular mechanism(s) behind such changes. The production of MVs is increased in cancer or damaged cells possibly due to the metabolic re-programming of these cells. Are there any data showing a direct connection between MTs and MVs formation? Or is this an indirect effect caused by the changes in the cell’s program?
How knockdown of beta3 tub affects MT dynamics in other cell types (discussion section – e.g. Neuronal-Specific TUBB3 Is Not Required for Normal Neuronal Function but Is Essential for Timely Axon Regeneration. Latremoliere A, et al Cell Rep. 2018 Aug 14;24(7):1865-1879.e9)
In my opinion, this project requires more experimental work. Also, the discussion has to be significantly improved.
Comments
If MTs are formed and beta3 tub is reduced, which isoform of beta-tubulin substituted beta3-tub?
Fig 1. Please provide large immunofluorescence images of the control and knockdown cells so one could clearly see changes.
Moreover, cells should be stained with for example an anti-a tubulin antibody to show if there are changes in the entire microtubular network in beta3-tub siRNA treated cells.
Fig1F – this image also is too small and thus not informative.
Figure 2 is missing (?)
According to Shibazaki et al., 2012, in in vitro cultured cancer cells, constitutive TUBB3 expression seems to be regulated in a cell-cycle dependent way, with maximal expression at the G2/M phase of the cell cycle [Shibazaki M, et al., Transcriptional and post-transcriptional regulation of betaIII-tubulin protein expression in relation with cell cycle-dependent regulation of tumor cells. International journal of oncology 2012, 40(3):695–702.]. Thus it would be of interest to investigate how knockdown of beta3 tubulin affects cell cycle (proliferation time, spindle morphology).
According to the Authors (line 135) beta 3 tub has a unique molecular structure. Beta 3 tubulin differ from other beta tubulins by several amino acids in the globular part (1-429, including a region of the paclitaxel binding domain) and by the C-terminal end (430-450) the place of the several tubulin posttranslational modifications. How knockdown of Beta3-tub changed resistance to paclitaxel and level of tubulin posttranslational modifications (e.g. acetylation and glutamylation?)
Minor comments:
Line 34: cellular microtubules – simply: microtubules
Line 43: cite newer review e.g. Latifkar et al., 2019 J Cell Sci 132
Line 75: perhaps change the phrase “MV(s) are important” for melanoma
Line 137: For example, β3-tubulin expressing Hela and MCF-7 cells exhibited more resistance to paclitaxel, a..
Shouldn’t be: For example, Hela and MCF-7 cells expressing β3-tubulin exhibited more resistance to paclitaxel, a…
Author Response
Reviewer: 2
Reviewer #2: The submitted manuscript “Beta3-tubulin knockdown suppresses microtubule dynamics and decreases the spontaneous release of microvesicles in human malignant melanoma cells (A375)” provides rather preliminary data. It shortly describes simple observations that knockdown of beta3 tubulin changes dynamics of the microtubules and that such cells release fewer microvesicles. Regretfully, the Authors failed to provide a hypothesis of the molecular mechanism(s) behind such changes. The production of MVs is increased in cancer or damaged cells possibly due to the metabolic re-programming of these cells. Are there any data showing a direct connection between MTs and MVs formation? Or is this an indirect effect caused by the changes in the cell’s program?
Authors response: The authors have conducted additional mechanistic studies. Despite the importance of β-tubulin isotypes in resistance to antimitotic drugs, there are no studies to date which examine the role of b3-tubulin in malignant melanoma. Herein, the authors use flow cytometry to investigate the effect of β3-tubulin siRNA on cell cycle distribution in A375 melanoma cells. The robust induction of G2/M arrest and the significant increase in polyploidy of β3-tubulin siRNA-treated cells provides direct evidence that β3-tubulin alters melanoma cell cycle regulation. Furthermore, G2/M arrest, which may be the result of defective mitotic spindle formation, enhances the cytotoxic effect of chemotherapy in melanoma cell lines.
How knockdown of beta3 tub affects MT dynamics in other cell types (discussion section – e.g. Neuronal-SpecificTUBB3 Is Not Required for Normal Neuronal Function but Is Essential for Timely Axon Regeneration. Latremoliere A, et al Cell Rep. 2018 Aug 14;24(7):1865-1879.e9). In my opinion, this project requires more experimental work. Also, the discussion has to be significantly improved.
Authors response: The authors have added a paragraph in the discussion section to further address the comments by Reviewer #2.
Comments
Reviewer #2: If MTs are formed and beta3 tub is reduced, which isoform of beta-tubulin substituted beta3-tub?
Authors response: As β3-tubulin is one of the major MT structural proteins, its relative reduction significantly altered the MTs assembly and disassembly processes (elongation and shortening). These data addressed our hypothesis that β3-tubulin mediates MT dynamicity and ultimately reduces the cell’s capacity to release MV. Furthermore, β3-tubulin siRNA-treated A375 cells were still expressing the protein (see immunofluorescence, Figure 1B & Supplemental figure S1) as the siRNA would not completely knockout β3-tubulin expression, so that we would be able to detect it, which may explain the continuation of the formation of, somewhat less functional, MT.
Reviewer #2: Fig 1. Please provide large immunofluorescence images of the control and knockdown cells so one could clearly see changes.
Authors response: Larger immunofluorescence images of the control and knockdown cells were added to the supplementary data file (Supplemental figure S1).
Reviewer #2: Moreover, cells should be stained with for example an anti-a tubulin antibody to show if there are changes in the entire microtubular network in β3-tubulin siRNA treated cells.
Authors response: Microtubule-associated proteins, which have been extensively studied, serve to link microtubules to each other and to other organelles [Cyr, 1991; Hirokawa, 1994; and Mandelkow, 1995]. Rather than using an anti-a tubulin antibody to determine the effects of β3-tubulin knockdown on MT dynamics, the authors transfected A375 cells with EGFP-microtubule-associated protein-4 cDNA (EGFP-MAP4). Please refer to page no. 3, line no. 85; page no. 5, line no. 115; page no. 9, line no. 216; and figure no. 2A. The microtubular network was visualized as shown.
Reviewer #2: Fig1F – this image also is too small and thus not informative.
Authors response: The image was removed from the figure.
Reviewer #2: Figure 2 is missing (?)
Authors response: The authors believe that the absence of figure 2 from the main text was due to a technical error. They have sent a note to the editorial assistant team to make sure that all figures are included in the text.
Reviewer #2: According to Shibazaki et al., 2012, in in vitro cultured cancer cells, constitutive TUBB3 expression seems to be regulated in a cell-cycle dependent way, with maximal expression at the G2/M phase of the cell cycle [Shibazaki M, et al., Transcriptional and post-transcriptional regulation of beta III-tubulin protein expression in relation with cell cycle-dependent regulation of tumor cells. International journal of oncology 2012, 40(3):695–702.]. Thus it would be of interest to investigate how knockdown of beta3 tubulin affects cell cycle (proliferation time, spindle morphology).
Authors response: The effect of β3-tubulin knockdown on the cell cycle regulation was studied. The results demonstrate that β3-tubulin knockdown induces G2/M arrest and a significant increase in polyploidy as compared to naïve- or control siRNA-treated cells.
Reviewer #2: According to the Authors (line 135) beta 3 tub has a unique molecular structure. Beta 3 tubulin differ from other beta tubulins by several amino acids in the globular part (1-429, including a region of the paclitaxel binding domain) and by the C-terminal end (430-450) the place of the several tubulin posttranslational modifications. How knockdown of Beta3-tub changed resistance to paclitaxel and level of tubulin posttranslational modifications (e.g. acetylation and glutamylation?)
Authors response: The data suggest that β3-tubulin knockdown suppresses microtubule dynamics and decreases the spontaneous release of microvesicles in human malignant melanoma cells (A375). To highlight the importance of β3-tubulin in mediating MT dynamics, cell migration, and drug resistance in other cancer cell types, the authors discuss previously published work that examines the relationship between β3-tubulin expression and resistance to paclitaxel. Though interesting and clinically relevant, the proposed additional studies are beyond the scope of this manuscript.
Minor comments:
Reviewer #2: Line 34: cellular microtubules – simply: microtubules
Authors response: Agreed and corrected.
Reviewer #2: Line 43: cite newer review e.g. Latifkar et al., 2019 J Cell Sci 132
Authors response: We thank the reviewer for bringing this new review to our attention. Our reference list has been updated accordingly.
Reviewer #2: Line 75: perhaps change the phrase “MV(s) are important” for melanoma
Authors response: Agreed and corrected
Reviewer #2: Line 137: For example, β3-tubulin expressing Hela and MCF-7 cells exhibited more resistance to paclitaxel, a..Shouldn’t be: For example, Hela and MCF-7 cells expressing β3-tubulin exhibited more resistance to paclitaxel, a…
Authors response: Agreed and corrected.